# Improving the availability of antihypertensive drugs in the India Hypertension Control Initiative, India, 2019–2020

**Abhishek Kunwar[1], Prabhdeep Kaur [2]\*, Kiran Durgad[1], Ganeshkumar Parasuraman[2], Meenakshi Sharma[3], Sudhir Gupta[4], Balram Bhargava[3], on behalf of "India Hypertension Control Initiative (IHCI) collaboration[¶]**

**1** Dept of Noncommunicable Diseases, WHO Country Office for India, New Delhi, India, **2** Division of Noncommunicable Diseases, ICMR-National Institute of Epidemiology, Chennai, India, **3** Indian Council of Medical Research (ICMR), New Delhi, India, **4** Directorate General of Health Services, Ministry and Health, and Family Welfare, New Delhi, India

¶ Membership of the India Hypertension Control Initiative (IHCI) collaboration is listed in the Acknowledgments.
\* kprabhdeep@gmail.com

## Abstract

### Background

Antihypertensive drug supply is sometimes inadequate in public sector health facilities in India. One of the core strategies of the India Hypertension Control Initiative (IHCI) is to improve the availability of antihypertensive drugs in primary and secondary care facilities. We quantified the availability of antihypertensive drugs in 2019–20 and described the practices in supply chain management in 22 districts across four states of India.

### Methods

Twenty-two districts from 4 states (Punjab, Madhya Pradesh, Telangana, and Maharashtra) were studied. We described the practices and challenges in supply chain management. We collected data on drug procurement from 2018 to 2020 and drug availability from April 2019 to March 2020. Quantity procured, the proportion of facilities with stockout at the end of each quarter, and availability of drugs in patient days were tabulated.

### Results

All states selected drug- and dose-specific protocols with Amlodipine as the initial drug and shifted to morbidity-based forecasting. The total number of antihypertensive tablets procured for the 22 districts increased from 16 million in 2017–2018 to 160 million in 2019–2020. The proportion of facilities with Amlodipine stock-out was below 5% during the study period. Amlodipine stock was available for at least 60 patient days from the third quarter of 2019 onward in all districts.

### Conclusions

This study demonstrates that including best practices can gradually strengthen the procurement and supply chain for antihypertensives in a low-resource setting. As the program was

**Data Availability Statement:** Kaur, Prabhdeep (2023), "Availability of antihypertensive drugs in project districts - India Hypertension Control

Initiative", Mendeley Data, V2, doi: 10.17632/
xs8dpz2z6v.2.

**Funding:** Source of funding Indian Council of
Medical Research, New Delhi project ID No.50/2/
TF-CVD/2019-NCD-I Role of funder: Sponsor
constituted a group of independent experts who
peer-reviewed the design and implementation of
the project.

**Competing interests:** The authors have declared
that no competing interests exist.

rapidly growing, there were still gaps in the procurement and distribution system which
needed to be addressed to ensure the adequacy of drugs. We recommend that best prac-
tices, including choosing a single protocol, basing supply on projected patient load rather
than an increment from historical levels, and using simple stock management tools, be repli-
cated in other districts in India to increase and sustain coverage of hypertension treatment.

## 1. Introduction

In India, there are estimated to be more than 200 million adults living with hypertension [1].
However, less than half of them are aware of their hypertension status, and less than one-tenth
of all people with hypertension have their blood pressure under control [2, 3]. Most low- and
low-middle-income countries have insufficient availability of antihypertensive medications
[4]. Non-availability of blood pressure-lowering medications is strongly associated with poor
control of hypertension [5]. In India, in a survey conducted across six states in 2004–05, the
median availability of 27 essential medicines in the public sector ranged from 0–30% [6]. A
study conducted in public health facilities of two North Indian states found that the availability
of antihypertensive drugs was only 60%. [7] A survey from Kerala, a southern state with a
good primary healthcare system, reported that the availability of cardiovascular disease and
diabetes medicines in the public sector health facilities was below the 80% target. [8]A
national-level survey reported inadequate drugs for noncommunicable diseases in primary
care facilities [9].

The India Hypertension Control Initiative (IHCI), launched in November 2017, aims to
reduce premature cardiovascular deaths by strengthening hypertension management and con-
trol at primary and secondary public healthcare facilities. The initiative includes a standardized
approach drawn from the World Health Organization (WHO) HEARTS technical package for
cardiovascular disease management in primary health care [10]. Adopting standard, drug- and
dose-specific hypertension treatment protocols and ensuring an uninterrupted supply of the
selected protocol medications are critical components of this initiative. The project was imple-
mented in the public sector health facilities of 26 districts in five states from 2018–19. The proj-
ect team worked closely with the state governments to improve the availability of
antihypertensive drugs at the district and facility levels.

Improving access to essential medicines requires appropriate selection, financing, afford-
able prices, and supply systems. [11] Several interventions were initiated to improve the selec-
tion, procurement, and availability of antihypertensive drugs in the public sector health
facilities implementing IHCI in 2018–19. We monitored the availability of drugs in these facili-
ties in 2019–20 and worked with stakeholders to understand the progress and challenges in
improving the availability of the protocol drugs. This paper describes practices in supply chain
management, the number of drugs procured and drug availability at the facility level in 2019–
20 in the four Indian states implementing the IHCI project.

## 2. Methods

### 2.1. Design and setting

The study included all 22 districts from 4 states (Punjab, Madhya Pradesh, Telangana, and
Maharashtra) implementing the IHCI. We documented the drug availability and challenges
experienced in improving drug availability at various levels from state-level procurement

agencies and regional and district warehouses in the project districts. We considered only public sector facilities, including primary and secondary care facilities. Most districts had one secondary care district hospital.

We collected data regarding drug procurement from 2018 to 2020 and drug availability at district and health facilities from April 2019 to March 2020 in the 22 districts. We estimated the availability of antihypertensive drugs in the standard treatment protocol at the health facility and district level, post-intervention in 2019–20, in 22 districts across four states of India.

## 2.2 Description of interventions

The project has five core strategies: protocol-based care, availability of adequate drugs, task sharing, patient-centered care, and a cohort-based monitoring system. Under IHCI, particular focus is rendered to ensure the uninterrupted availability of drugs at all service delivery points. The project aimed to ensure the timely purchase of adequate quantities of hypertension drugs, equitable distribution of hypertension protocol drugs to service delivery points, and strengthening systems for monitoring and timely action. The project team developed multiple easy-to-adopt tools, curated interventions for improving supply chain efficiency, and extended continuous technical support. The critical interventions implemented from 2018 onwards to streamline the supply chain of antihypertensives were as follows:

**Selection of drugs.**   We focused on procuring and supplying a limited number of antihypertensive drugs in line with the adopted treatment protocol (S1 and S2 Figs). This approach helped streamline forecasting, efficient procurement with affordable drug costs, and improved stockkeeping and distribution.

**Forecasting drug requirements and budgeting.**   Before the IHCI, the traditional method of forecasting drug requirements was to project future consumption by adding a small percentage increase to past consumption. This approach is not suitable for scaling up programs. The morbidity-based method, by contrast, accounts for future program growth, but this method was not practised before the IHCI due to the complexity of calculations required and lack of training. The morbidity-based method considers expected patients to be initiated on treatment in addition to patients who are already on treatment and the drugs in the protocol at various levels. We developed a customized, easy-to-use forecasting tool (S1, S2 Files). Based on the forecasted drug requirement, assistance was extended for budget planning and prioritizing through rationalization of existing funds and mobilization of additional funding through the national program and/or state funds [12].

**Procurement.**   One of the most critical interventions is timely tendering before the expiry of the existing rate contract to ensure continuity of rate contract availability. We recommended a multi-year rate contract instead of annual tendering to reduce the administrative processes and bottlenecks. We also suggested multiple supplier empanelments to ensure the availability of an alternative supply source in case of failure/non-compliance by any one supplier. Many states issued the procurement order for the entire year; however, there were challenges for the supplier and storage at the warehouse level. One of the more pragmatic strategies was to purchase orders with scheduled supply or scheduled procurement, which helped maintain adequate drug availability with minimal inventory holding. This approach helped mitigate the storage space challenge and minimized the risk of drug expiry and product deterioration during storage. The scheduled supply approach also acted as assurance for the supplier and proved helpful in planning and executing orders in time. Although most of the procurement was done at the state level, districts were provided funds and empowered to stopgap local procurement if there was a disruption in the procurement/supply of drugs through an established state mechanism.

**Storage and distribution.**   Every facility was advised to maintain 2–3 months' stock based on the total number of registered patients. Ready reckoners were developed to quickly assess stock position for patient load-linked indent/drug distribution and periodic (monthly) refilling (S3, S4 Files)). The health facility or district store arranged vehicles to deliver drugs to service delivery points from the issuing stores.

**Dispensing and monitoring.**   All states issued advisories to the districts to dispense at least 30 days of drugs per treated patient and monitor the stocks. We used easy-to-use and straightforward formats/tools for pharmacists and healthcare providers for inventory recording and reporting (S5 File). Health facilities with suboptimal stock levels were monitored monthly at the district, state, and national levels with a feedback mechanism to drive prompt supply chain action to fill in the gaps.

**Training and capacity building.**   We trained all pharmacists and program managers on the treatment protocol, annual requirement forecasting, monthly/periodic indent/stock refill based on the patients under treatment, stock level monitoring, and inventory records.

## 2.3 Data collection

**Review of documents.**   We reviewed essential drug lists (EDLs), tender documents, procurement policy, past procurement data, drug distribution guidelines, and other relevant documents in the state-level procurement agencies, state NCD program unit, and district warehouses. The documents provided information regarding existing forecasting, procurement, and distribution practices.

**Field-based observations.**   The IHCI intervention team of public health specialists supported the state and district-level NCD program units. The team observed the practices regarding existing supply chain processes from procurement to dispensing to identify the challenges and to support streamlining the process.

**Quantitative data for drugs.**   We collected data on the annual procurement of all antihypertensive drugs for three financial years, i.e., 2017–18, 2018–19, and 2019–20, from state and district records.

We also collected data on hypertension protocol drugs stock throughout the study from all public sector health facilities of these districts at the end of each quarter for four quarters [from quarter 2 (Q2) 2019 –quarter 1 (Q1) 2020] from stock ledgers and logistics management information system (LMIS) software. The quarterly drug stock reporting also consisted of a cumulative number of patients registered at the health facility under the IHCI to estimate the drug stock in patient days. These data were extracted from the Simple digital hypertension management app in Maharashtra and Punjab and IHCI paper-based registers in Madhya Pradesh and Telangana.

## 2.4 Data analysis

We used the "Logistics cycle model" to identify the thematic areas for analysis. [13] We summarised various aspects of the supply chain, from selecting antihypertensive drugs to dispensing the drugs at the facility level.

*a. Qualitative analysis*

We analyzed and summarized policy documents, procurement records, monthly/annual reports, and field observations in five thematic areas (Fig 1).

*b. Quantitative analysis*

We analyzed three indicators using state, district, and health facility level data to assess if the interventions improved the availability of antihypertensive drugs.

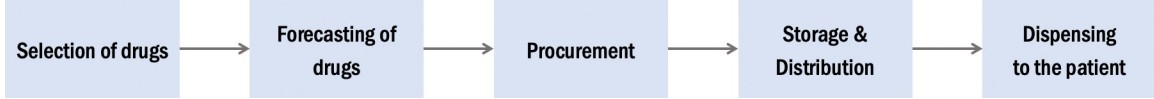

**Fig 1. Steps for antihypertensive medication supply chain management in the India Hypertension Control Initiative.**

i. Quantity of antihypertensive drugs procured for three financial years (2017–18, 2018–19, and 2019–20)

ii. The proportion of health facilities with drug stock out—Out of all implementing facilities, the number of facilities had nil stock of any protocol drugs at the end of each of the four quarters from Q2, 2019 to Q1, 2020. Stock out was separately considered for each of the three protocol drugs.

iii. Drug stock in patient-days at the end of each quarter at the district level for the four quarters from Q2, 2019 to Q1, 2020.

'Drug stock in patient-days' is the unit to measure how many days the available stock of a drug would last considering the number of patients under treatment and the average daily requirement of the drug per patient. The stock in patient-days for a drug is computed as 'Stock available in a number of tablets' / (Total number of patients x Number of tablets required per day).

Under IHCI, each state adopted a drug and dose-specific stepwise hypertension treatment protocol. The program adopted initial assumptions on the proportion of patients started on treatment who are controlled at each step of the protocol. [14] Based on the treatment protocol adopted and the assumed proportion of patients reaching blood pressure control at each treatment protocol step, the approximate requirement of each protocol drug per patient per month was calculated for two types of protocols (S3 File).

## 2.5 Human subjects protection

The ICMR-National Institute of Epidemiology, Chennai, institutional ethics committee approved the project. The study did not include any patient-level data. We collected the data and reviewed documents following the required administrative approval process.

## 3. Results

### 3.1 Selection of drugs

Public procurement is done for drugs listed in the state's Essential Drugs List (EDL). Within each therapeutic category, several drugs are listed and classified by the health facility's level (primary, secondary, or tertiary) where the drugs are to be made available. All four states had developed hypertension treatment protocols as one of the strategies under the IHCI (Table 1). [15] The states had included the protocol drugs in the state-specific EDL. Before IHCI, antihypertensive drugs were mostly procured for secondary and tertiary health facilities. After IHCI implementation, all protocol drugs were included in the EDL for primary, secondary and tertiary care facilities.

### 3.2 Forecasting drug requirements

Requirement forecasting is a crucial step in the supply chain to ensure the availability of adequate drugs. In all four states, essential drug requirements were collected from the district level annually for procurement and budget planning. The facilities submit the requirement to the

**Table 1. Drug selection, forecasting, procurement, storage and distribution practices for antihypertensive drugs in four Indian states, 2020.**

| Components | Maharashtra | Punjab | Madhya Pradesh | Telangana |
|---|---|---|---|---|
| **Selection of drugs** | | | | |
| Availability of Essential Drug List | Yes | Yes | Yes | Yes |
| Protocol drugs in essential drugs list | Amlodipine, Telmisartan, Chlorthalidone | Amlodipine, Telmisartan, Chlorthalidone | Amlodipine, Telmisartan, Hydrochlorothiazide | Amlodipine, Telmisartan, Hydrochlorothiazide |
| Availability of protocol drugs at various health facility levels | All levels | All levels | All levels | All levels |
| **Forecasting** | | | | |
| Frequency of annual demand generation | Yearly | Yearly | Yearly | Yearly |
| Method of drug forecasting | IHCI tool–Morbidity method | IHCI tool–Morbidity method | IHCI tool–Morbidity method | IHCI tool–Morbidity method |
| Review of indents | Yes | Yes | Yes | Yes |
| **Procurement** | | | | |
| Name of nodal agency | Haffkine Bio-Pharmaceuticals Corp. Ltd. | Punjab Health Systems Corporation | Madhya Pradesh Public health Service Corporation Limited | Telangana State Medical Services & Infrastructure Development Corporation |
| Rate contract period | Quantity contract | 2 years | 2 years | 2 years |
| Purchase order system | Centralized | Centralized | Decentralized | Centralized |
| Frequency of purchase order | Annual | Quarterly | Quarterly | Annual |
| Supply period | 45 days | 40 days | 45 days | 75 days |
| **Storage and distribution** | | | | |
| Primary stores | District stores | Regional stores | District stores | Regional stores |
| Responsible for transportation from primary stores | Health facilities | Health facilities | Health facilities | Health facilities |
| Frequency of stock replenishment at health facilities | Monthly | Monthly | Monthly | Monthly |

district, and a consolidated report is submitted to the state. The annual requirements of antihypertensive drugs were estimated based on last year's consumption until 2017–18. The project team worked with the state-level stakeholders to estimate the requirement of the drug-using morbidity-based forecasting in 2018–19 and 2019–20 (Table 1). To forecast the annual drug requirement based on the total number of patients under care, expected new enrolment over the year, and the adopted treatment protocol, all 22 districts started using the forecasting tool developed for the project (S1, S2 Files).

### 3.3 Procurement

All four states had fully functional state-owned procurement agencies (Table 1). The drug rate contract was renewed every two years in all states except Maharashtra, where a quantity contract was used. The drug purchase order system was centralized in all states except Madhya Pradesh. In Madhya Pradesh, districts were authorized to purchase from manufacturers directly. Annual purchase orders were placed in Maharashtra and Telangana, whereas quarterly in Madhya Pradesh and Punjab.

All state procurement agencies procured drugs directly from the manufacturers. A good manufacturing practices compliance certificate and production experience of at least three years were mandatory to procure the drugs from credible sources. Although the manufacturers mandatorily do quality control tests before supplying stocks to the states, all state procuring

agencies retest the samples at independent empanelled testing laboratories to confirm the quality of the drug before the batches are released to the health facilities.

IHCI field teams observed the key challenges which led to delays in procurement were non-availability of rate contracts, lack of multiple suppliers, delays in tendering and placing purchase orders, delayed or partial supply by vendors, delays in receiving funds and lack of well-defined budget heads for antihypertensive drugs. As per procurement data, the procurement improved in 2019–20, but several challenges could not be addressed. The effective interventions at the procurement level were close monitoring and follow-up of tenders by state NCD nodal units, providing reliable estimates by forecasting, the collaboration between procurement cell and NCD unit for purchase scheduling, accelerating the quality control process to reduce quarantine time, and strategic procurement when contract nears expiry. At the district level, local procurements helped fill the gaps when the supply from the state was delayed.

All states increased the quantities of antihypertensive drugs procured for the project districts after the project's initiation (Fig 2). In Madhya Pradesh, there was a three-fold increase in the procurement of amlodipine and telmisartan in 2018–19 and a six-fold increase in the procurement of diuretics. The procurement of amlodipine was 3-10-fold higher in 2019–20 compared to 2017–18 in Telangana, Maharashtra and Punjab. Punjab and Maharashtra did not procure telmisartan in 2017–18. Telmisartan was procured in small quantities in Punjab in 2018–19, and nearly 4 million tablets were procured in 2019–20. The procurement of telmisartan in Telangana was lower in 2018–19 compared to the previous year due to manufacturers' supply delays. In Maharashtra, delays in the tender process led to lower procurement in 2019–20 compared to the previous year. The other states did not procure diuretics except for Madhya Pradesh in 2017–18. Initially, the other three states had to add chlorthalidone or hydrochlorothiazide in the EML, and procurement was initiated in the subsequent years. The total number of tablets procured for all antihypertensive medications for the 22 districts in the four states increased from 16 million tablets in 2017–2018 to 160 million tablets in 2019–2020 (Fig 3).

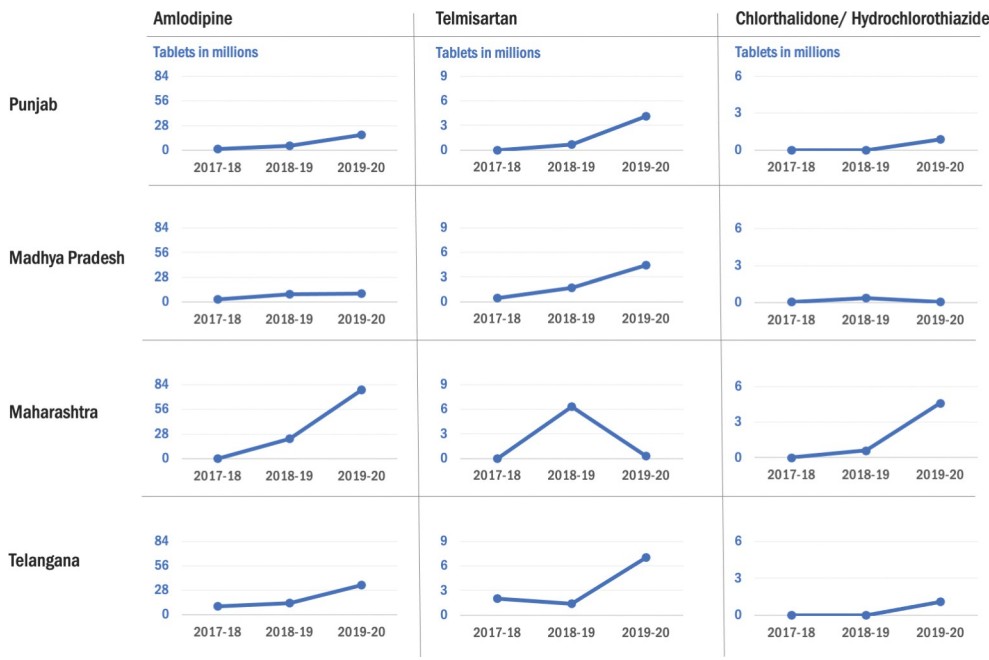

**Fig 2. Volume of procurement of antihypertensive drugs (in millions of tablets) by type of drug and by state for 2017–18, 2018–19, 2019–20.**

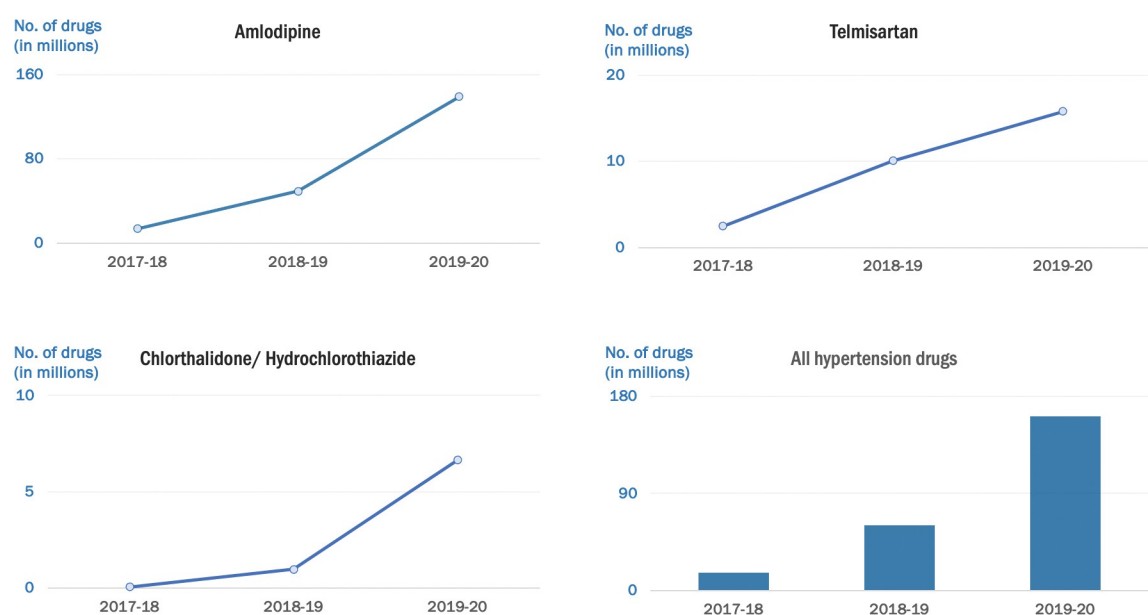

**Fig 3. Volume of procurement of all antihypertensive drugs (in millions of tablets) drugs overall and by type of drug for all study districts for 2017–18, 2018–19, 2019–20.**

### 3.4 Storage and distribution

Drugs are directly delivered to the district stores based on the purchase orders in Maharashtra and Madhya Pradesh. The district stores catered to the health facilities. In Punjab and Telangana, regional drug warehouses received the drugs from the suppliers and further distributed them to the district stores, which supplied all health facilities in the district.

At the facility level, stocks were replenished monthly (Table 1). The refill quantity was assessed using ready reckoners (S3, S4 Files). The monthly cycle helped in overcoming the storage-related challenges at the facility level. The pharmacists from health facilities sent an indent, arranged transport, and collected drugs from the primary/district levels stores, usually once a month. The distribution of drugs to the health facilities is based on a requisition or pull system, and the quantity to be issued is based on requests placed by the health facilities. Current practices utilise ambulances and other program monitoring vehicles available at health facilities to pick up the drug stocks. Thus, the availability of drugs is affected by the availability of the vehicle and can lead to a shortage and stockout of drugs at health facilities even when there are adequate stocks at the district/issuing stores. Given these constraints, district-level warehouses were easier to access than regional or state-level warehouses.

IHCI field teams observed three key challenges. The first was the availability of transport for collecting drugs from the district warehouses. The second challenge was a lack of understanding among pharmacists regarding estimating drug requirements for requisitions sent to the drug stores. The third challenge was the lack of rational distribution of drugs, which led to excess stocks in some facilities and stockouts in others. Under IHCI, min-max inventory levels were affixed for the health facilities. A ready reckoner was prepared for pharmacists to assess the antihypertensive drugs required for a given number of patients enrolled in the treatment (S4 File). The pharmacists reported that the ready reckoner helped prepare monthly requisitions and reduced stockouts.

### 3.5 Availability of drugs at the district level and point of care

The project began in 2018 and enrolled 68,982 patients in the first year and 283,474 in 2019. The total number of patients enrolled under IHCI in the 22 districts increased from 83,553 in Jan 2019 to 441,929 in March 2020.

Amlodipine stock was available for at least 60 patient days in all districts at the end of every quarter from Q2 2019 in Punjab, Maharashtra and Madhya Pradesh and August 2019 onwards in Telangana (Fig 4). Telmisartan/Losartan was available for 60 patient days or higher for all nine months in Punjab and Madhya Pradesh. The availability of Telmisartan/Losartan was below 60 patient days for two months (Jan 2019 and March 2019) in Maharashtra and four months (Dec 2019 to March 2020) in Telangana. Diuretics were supplied from August 2019 onwards in Telangana and were available for 120 patient days or higher after that. In

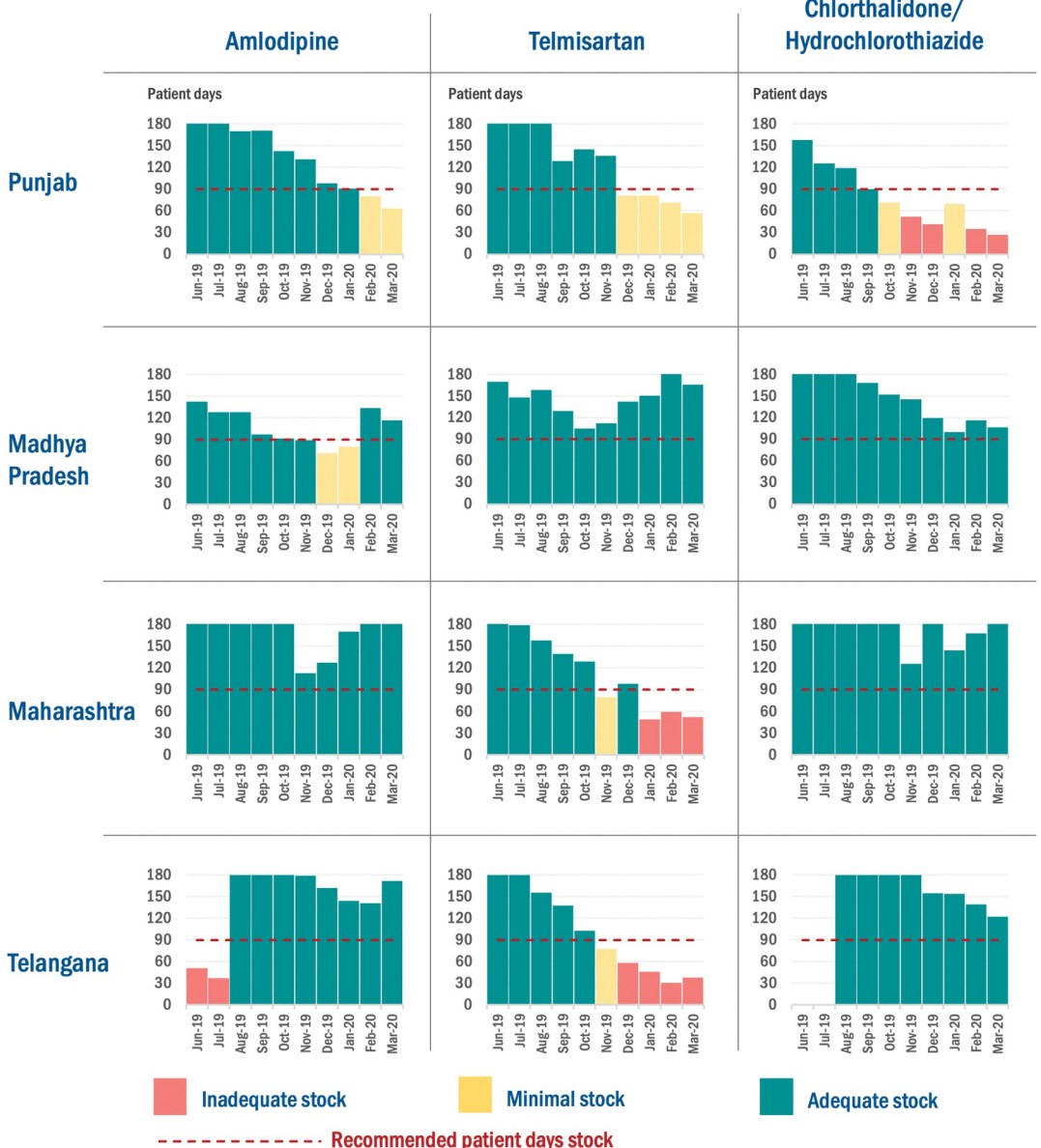

**Fig 4. Antihypertensive drug availability inpatient days by the state for four states of India, Jun 2019 –March 2020.**

**Table 2. Proportion of facilities with stockouts (no drugs available) at the end of each quarter, 2019–20.**

| | June 2019 | | | September 2019 | | | December 2019 | | | March 2020 | | |
|---|---|---|---|---|---|---|---|---|---|---|---|---|
| | Total No. of health facilities | No. of Facilities with stock out | % with stock out | Total No. of health facilities | No. of Facilities with stock out | % with stock out | Total No. of health facilities | No. of Facilities with stock out | % with stock out | Total No. of health facilities | No. of Facilities with stock out | % with stock out |
| **Amlodipine** | | | | | | | | | | | | |
| Punjab | 155 | 3 | 1.9 | 155 | 9 | 5.8 | 155 | 13 | 8.4 | 155 | 13 | 8.4 |
| Madhya Pradesh | 155 | 0 | 0.0 | 171 | 0 | 0.0 | 171 | 1 | 0.6 | 171 | 0 | 0.0 |
| Maharashtra | 218 | 2 | 0.9 | 218 | 3 | 1.4 | 218 | 8 | 3.7 | 218 | 3 | 1.4 |
| Telangana | 199 | 2 | 1.0 | 199 | 3 | 1.5 | 201 | 1 | 0.5 | 201 | 0 | 0.0 |
| **Telmisartan** | | | | | | | | | | | | |
| Punjab | 155 | 11 | 7.1 | 155 | 24 | 15.5 | 155 | 21 | 13.5 | 155 | 33 | 21.3 |
| Madhya Pradesh | 155 | 7 | 4.5 | 171 | 4 | 2.3 | 171 | 5 | 2.9 | 171 | 1 | 0.6 |
| Maharashtra | 218 | 5 | 2.3 | 218 | 4 | 1.8 | 218 | 13 | 6.0 | 218 | 23 | 10.6 |
| Telangana | 199 | 9 | 4.5 | 199 | 4 | 2.0 | 201 | 11 | 5.5 | 201 | 25 | 12.4 |
| **Diuretic** | | | | | | | | | | | | |
| Punjab | 155 | 106 | 68.4 | 155 | 93 | 60.0 | 155 | 117 | 75.5 | 155 | 155 | 100.0 |
| Madhya Pradesh | 155 | 66 | 42.6 | 171 | 17 | 9.9 | 171 | 10 | 5.8 | 171 | 19 | 11.1 |
| Maharashtra | 218 | 7 | 3.2 | 218 | 21 | 9.6 | 218 | 30 | 13.8 | 218 | 47 | 21.6 |
| Telangana | 199 | 199 | 100.0 | 199 | 65 | 32.7 | 201 | 11 | 5.5 | 201 | 14 | 7.0 |

Maharashtra and Madhya Pradesh, diuretics were available for 100 patient days or higher throughout nine months (Fig 4).

The proportion of facilities with amlodipine stock out was below 5% in all states during the study period (Table 2). As the program was rapidly expanding, Punjab and Maharashtra had an increase in the facilities with stockout in the last quarter of 2019. Less than 10% of facilities had Telmisartan stockouts between June 2019 and Dec 2020. In the first quarter of 2020, three states except Madhya Pradesh had more than 10% of facilities with stockouts for telmisartan/losartan. Stockouts were higher for diuretics compared to the other two drugs.

## 3.6 Dispensing to patient

Before the IHCI project implementation, the number of days drugs were dispensed varied from seven to 30 days, depending on the availability of drugs. After initiating IHCI, all four states issued guidelines, and medical officers and pharmacists were trained to prescribe and dispense the antihypertensive drugs for 30 days.

## 3.7 Financing and Logistic Management Information Systems

In India, all states procured drugs by pooling funds from the national level and state-level [12]. The annual budget is prepared by the states and submitted to the state and central government. All four states have digital logistic management information systems (LMIS) for managing drug procurement, inventory, distribution and monitoring. This system has been used in Madhya Pradesh and Telangana since 2014 and Maharashtra and Punjab since 2017 at all levels, from the state to the primary health care level. The institutional mechanism was in place for monthly drug stock reviews in all four states. Regular monitoring and supervisory visits were done to all types of drug stores, and records were maintained with respective supervisors.

All state pharmacists used digital platform reports for periodic review and appropriate action. However, none of the district officials used digital platform reports.

## 4. Discussion

We assessed the functioning of the antihypertensive drugs supply chain steps from selection, forecasting practices, and procurement to availability at the point of care delivery in 22 districts over the beginning phase of IHCI implementation (2018–2020). Overall, procurements of antihypertensive protocol drugs increased to volumes closer to patient-day needs for the growing program, and their availability improved at the primary health care level. States planning to scale up hypertension control programs should adopt a package of interventions including drug-dose specific treatment protocol, forecasting using tools, multi-year rate contracts, a ready reckoner at the facility level and key indicators for monitoring drug availability. The interventions such as drug-dose specific treatment protocol, multi-year rate contracts and monthly monitoring of drug availability at the facility level are relevant for all disease conditions. However, forecasting tools and ready reckoners are useful for any chronic disease program where patients need to take drugs on a long-term basis.

WHO recommends rational selection of essential drugs as a critical strategy for improving treatment access. [11] The IHCI project team worked with stakeholders to develop simple drug and dose-specific protocols [15]. The protocols were consistent with the recent WHO hypertension treatment guidelines [16]. Globally, many countries have developed drug-specific protocols for hypertension treatment [17]. Before implementing the IHCI, there was no uniformity in the prescription practices, making drug procurement challenging. The protocols are beneficial for program managers to plan procurement. Program managers need to procure fewer drugs, each in larger quantities, which pools demand and leads to an economy of scale, thereby reducing the unit cost of the drugs. The states procured only three drugs in the project districts for the primary care facilities, making the supply chain more efficient.

One of the significant challenges in the supply chain is the lack of accurate estimation of drug requirements at the facility and district levels. Procurement of insufficient quantities of medicines due to poor forecasting and unexpected demand challenges has been documented in the LMIC health systems [18]. Before implementing IHCI, the estimated requirements were mainly based on the previous year's consumption. The consumption-based approach underestimates the volume required if the facility has repeated stockouts or the program is expanding and treating more patients. Under the IHCI, hypertension protocol-based annual drug forecasting based on the existing patient load and additional expected patient load made the estimates more reliable and reduced stockouts. Estimating periodic stock replenishment at health facilities requires sound information systems that track the number of patients on treatment rather than the number of visits to the health centre. Prior to implementing IHCI, only the aggregate data on the number of patient visits was collected and reported, but there was a lack of a registry. Once the information system was in place, ready reckoners based on patient registration helped pharmacists estimate monthly demand. The morbidity-based estimation method will be useful until the enrolment of a large proportion of patients expected to take treatment in the public sector. Once the district has achieved saturation based on utilization of public sector health facilities, the transition to the consumption-based method can be considered based on the assessment.

In the past decade, most Indian states have developed a centralized procurement mechanism under a dedicated procurement agency at the state level. To implement the interventions mentioned in this study, the states should have a department/agency that oversees the procurement process, a well-defined procurement policy to govern supplier selection and contract management, a coordination mechanism between the user department and procurement unit,

a risk mitigation plan for supply/quality failures and timely payment mechanism. Procurement improved with better planning, continuous follow-up with various stakeholders, and forecasting in the study districts. As measured by patient days, the overall availability of the drugs was adequate in the study period, and there were few stockouts for the first two protocol drugs. A systematic review of studies from LMICs reported that procurement challenges included insufficient funds, delays in procurement, and insufficient procurement due to poor forecasting [18]. Robust governance, efficiency and transparency in procurement and contracting practices are essential to improve drug availability. We could not fully document and address these challenges in our study. We need more in-depth studies to explore these challenges and better coordination between the program managers and procurement teams to reduce delays.

Despite challenges, at least two protocol drugs were available throughout the study period. Several studies from India and other LMICs have documented poor availability of essential drugs. [7, 8, 19–22] A national-level survey of public and private sector facilities reported poor NCD drugs and technologies in primary care. In the districts where the national NCD program was implemented, less than 1% of community health centres (which cater to approximately 100,000 population) and 17% of district hospitals were fully equipped with medicines and technologies to manage hypertension and cardiovascular diseases [9]. In Punjab, a survey of health facilities reported a 60% availability of antihypertensive drugs in 2015 [7]. Our data indicated an improvement in the availability of drugs in Punjab compared to 2015. Our experience reinforces the importance of building capacity and implementing good supply chain management practices to improve availability and reduce stockouts.

The limitation of our study was that we could not compare our data with the district-level availability of drugs before the start of the project. We used the procurement data as a surrogate for the availability of drugs in 2017–18 and 2018–19. We did not design a separate study for supply chain interventions. The lack of drugs was a major challenge in the initial planning and implementation phase. Hence, we planned the supply chain interventions described in the manuscript. Our experience can offer insights to researchers and program managers in low-resource settings. It can be used to design more structured, well-designed studies for supply chain interventions to improve the availability of NCD drugs in the context of LMIC.

We demonstrated the feasibility of strengthening procurement and supply chain management for antihypertensive drugs in a low-resource setting at scale. Although the number of procured drugs increased, there were still insufficient stocks at various time points. It was due to the challenges which could not be fully addressed, including timely tendering to ensure rate contracts, timely placement of orders, robust stock tracking system at the facility level, and timely availability of transport. It is important to address these gaps as the program expands and the number of patients on treatment increases. We were able to document improvement in several important areas, including the adoption of the treatment protocol, increased procurement volumes due to forecasting and collaboration among various stakeholders to match requirements with program scale-up, monthly stock monitoring at the facility level, improved availability of antihypertensive drugs included in the protocol in primary care facilities and minimum 30 days refills for the patients. Effective practices established in the IHCI can be replicated in other districts in India and other LMICs to increase the coverage of hypertension treatment. The interventions to further strengthen the governance and adoption of best procurement practices should be sustained to ensure the availability of drugs for NCDs.

## Supporting information

**S1 Fig. Sample treatment protocol 1.**
(TIFF)

**S2 Fig. Sample treatment protocol 2.**
(TIFF)

**S1 File. Drugs forecasting tool for sample treatment protocol 1.**
(XLSX)

**S2 File. Drugs forecasting tool for sample treatment protocol 2.**
(XLSX)

**S3 File. Monthly drug requirement matrix for sample protocols 1 and 2.**
(XLSX)

**S4 File. Drug adequacy ready reckoner for sample treatment protocols 1 and 2.**
(XLSX)

**S5 File. Data collection format for the stock position of antihypertensive drugs inpatient days.**
(XLSX)

## Acknowledgments

We thank the field-level health care workers, nurses, doctors, and district-level health officials for providing services for managing hypertension as part of the IHCI. We thank Senior Treatment Supervisors for their role in capacity building and supportive supervision in the project districts. We thank the ICMR Task Force Chair, Dr. Ambuj Roy, and all other experts for their valuable inputs in designing and implementing the project.

## List of authors (alphabetical order) for "India Hypertension Control Initiative (IHCI) Collaboration."

Sampada D Bangar[5], Vishwajit Bharadwaj[6], Rupali Bharadwaj[7], Sailaja Bitragunta[1], Sreedhar Chintala[8], Tapas Chakma[9], Tejpalsinh A Chavan[10], Sunil Dar[11], Bidisha Das[12], RS Dhaliwal[3], Sandeep Singh Gill[13], Tanu Jain[4], Padmaja Jogewar[14], Chakshu Joshi[15], Abhishek Khanna[16], Suhas N Khedkar[17], Ashish Krishna[18], Navneet Kumar[19], Vijay Kumar[20], Madhavi M[21], Anupam K Pathni[18], Satyendra N Ponna[22], Sravan K Reddy[23], Swagata K Sahoo[18], Ashish Saxsena[24], Bhawna Sharma[18], Shweta Singh[16], Gurinder B Singh[13], Sunny Swarnkar[4], Jatin Thakkar[25], Fikru T Tullu[2], Mohammed Abdul Wassey[26], Amol B Wankhede[27]

[1] Division of Noncommunicable Diseases, ICMR-National Institute of Epidemiology, Chennai

[2] Dept of Noncommunicable Diseases, WHO Country Office for India, New Delhi

[3] Indian Council of Medical Research (ICMR), New Delhi

[4] Directorate General of Health Services, Ministry and Health, and Family Welfare, New Delhi

[5] ICMR-National AIDS Research Institute, Pune

[6] IHCI Project, District NCD Cell, Bhandara (Maharashtra), WHO-India.

[7] IHCI Project, District NCD Cell, Chhindwara (Madhya Pradesh), WHO-India.

[8] IHCI Project, District NCD Cell, Karimnagar (Telangana), WHO-India.

[9] ICMR-National Institute of Research in Tribal Health, Jabalpur

[10] IHCI Project, District NCD Cell, Sindudurg (Maharashtra), WHO-India.

[11] IHCI Project, District NCD Cell, Hoshiarpur (Punjab), WHO-India.

[12] IHCI Project, District NCD Cell, Bhatinda (Punjab), WHO-India.

[13] State NCD Cell, Department of Health and Family Welfare, Government of Punjab, Chandigarh

[14] State NCD Cell, Directorate of Health Services, Government of Maharashtra, Mumbai

[15] IHCI Project, District NCD Cell, Ratlam (Madhya Pradesh), WHO-India.

[16] NPCDCS Cell, DGHS, New Delhi, WHO- India

[17] IHCI Project, District NCD Cell, Satara (Maharashtra), WHO-India.

[18] Resolve to Save Lives, New Delhi

[19] IHCI Project, State NCD Cell, Chandigarh (Punjab), WHO-India

[20] IHCI Project, District NCD Cell, Gurdaspur (Punjab), WHO-India.

[21] State NCD Cell, Department of Health, Medical and Family Welfare, Government of Telangana, Hyderabad

[22] IHCI Project, District NCD Cell, Warangal (Telangana), WHO-India.

[23] IHCI Project, State NCD Cell, Hyderabad (Telangana), WHO-India.

[24] State NCD Cell, Directorate of Health Services, Government of Madhya Pradesh, Bhopal

[25] IHCI Project, State NCD Cell, Bhopal (Madhya Pradesh), WHO-India.

[26] IHCI Project, District NCD Cell, Mahabubnagar (Telangana), WHO-India.

[27] IHCI Project, State NCD Cell, Mumbai (Maharashtra), WHO-India.

## Author Contributions

**Conceptualization:** Abhishek Kunwar, Prabhdeep Kaur, Meenakshi Sharma, Sudhir Gupta, Balram Bhargava.

**Data curation:** Abhishek Kunwar, Prabhdeep Kaur, Kiran Durgad.

**Formal analysis:** Abhishek Kunwar, Kiran Durgad, Ganeshkumar Parasuraman.

**Funding acquisition:** Balram Bhargava.

**Methodology:** Prabhdeep Kaur, Meenakshi Sharma, Sudhir Gupta, Balram Bhargava.

**Project administration:** Sudhir Gupta.

**Software:** Ganeshkumar Parasuraman.

**Supervision:** Abhishek Kunwar, Prabhdeep Kaur, Kiran Durgad.

**Validation:** Ganeshkumar Parasuraman.

**Visualization:** Prabhdeep Kaur, Kiran Durgad, Ganeshkumar Parasuraman, Balram Bhargava.

**Writing – original draft:** Abhishek Kunwar, Prabhdeep Kaur, Kiran Durgad.

**Writing – review & editing:** Ganeshkumar Parasuraman, Meenakshi Sharma, Sudhir Gupta, Balram Bhargava.

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
