## [Decision Letter · Decision Letter 0]

15 Aug 2023

PONE-D-23-13904Improving the availability of antihypertensive drugs in the India Hypertension Control Initiative, India, 2019-2020PLOS ONE

Dear Dr. Kaur,

Thank you for submitting your manuscript to PLOS ONE. After careful consideration, we feel that it has merit but does not fully meet PLOS ONE’s publication criteria as it currently stands. Therefore, we invite you to submit a revised version of the manuscript that addresses the points raised during the review process. Please note Dr Satheesh's comments are given as comments on a PDF version of the manuscript, attached.

We look forward to receiving your revised manuscript.

Kind regards,

Dzintars Gotham

Academic Editor

PLOS ONE

Journal Requirements:

"NO authors have competing interests"

5. One of the noted authors is a group or consortium [India Hypertension Control Initiative (IHCI) Collaboration."]. In addition to naming the author group, please list the individual authors and affiliations within this group in the acknowledgments section of your manuscript. Please also indicate clearly a lead author for this group along with a contact email address.

7. We note that Figure 1 in your submission contain copyrighted images. All PLOS content is published under the Creative Commons Attribution License (CC BY 4.0), which means that the manuscript, images, and Supporting Information files will be freely available online, and any third party is permitted to access, download, copy, distribute, and use these materials in any way, even commercially, with proper attribution. For more information, see our copyright guidelines: http://journals.plos.org/plosone/s/licenses-and-copyright.

Reviewers' comments:

Reviewer's Responses to Questions

**Comments to the Author**

1. Is the manuscript technically sound, and do the data support the conclusions?

Reviewer #1: Yes

Reviewer #2: Yes

2. Has the statistical analysis been performed appropriately and rigorously? 

Reviewer #1: Yes

Reviewer #2: Yes

3. Have the authors made all data underlying the findings in their manuscript fully available?

Reviewer #1: Yes

Reviewer #2: Yes

4. Is the manuscript presented in an intelligible fashion and written in standard English?

Reviewer #1: Yes

Reviewer #2: Yes

5. Review Comments to the Author

Reviewer #1: Thank you for the opportunity to review this important piece of work, which demonstrates the feasibility of optimizing procurement and supply chains for antihypertensive drugs in India's resource-limited public sector. Please find my comments, questions, and suggestions on the attached proof of the manuscript.

Reviewer #2: The manuscript presents data and analysis that would be useful to readers, in particular health policymakers in India.

My main concern is that the conclusion and discussion of this paper is not reflective of the findings. It seems to try to gloss over procurement concerns that the data actually reveal. For example, consider the abstract's conclusions that "This study demonstrates the feasibility of strengthening the procurement and supply chain management for antihypertensive drugs." The data don't exactly demonstrate this. From table 2, I calculated that the overall percent of facilities with stockouts actually increased over the study period for both Amlo and Telmisartan. So even though the absolute quantity of pills procured did grow substantially, there were some inefficiencies still. According to figure 4, this seems to reflect issues with inadequate stock particularly of Telmisartan in 3 out of 4 examined states. For Amlo and Chlortalidone, issues with inadequate stock were observed in 1 out of 4 states.

Overall, the study demonstrates that the absolute quantities of pills did increase across the board, but some states still experienced inefficiencies with the sufficient procurement of some medicines, especially Telmisartan. The key conclusion of the paper should be to emphasize the need to resolve these stocking issues as scale up of the program increases. It's important to not bury the key take-away of the analysis just because it doesn't show perfect outcomes.

6. PLOS authors have the option to publish the peer review history of their article (what does this mean?). If published, this will include your full peer review and any attached files.

Reviewer #1: **Yes: **Gautam Satheesh

Reviewer #2: No

---

## [Author Response · Author response to Decision Letter 0]

12 Nov 2023

Reviewers' comments:

Reviewer #1: Thank you for the opportunity to review this important piece of work, which demonstrates the feasibility of optimizing procurement and supply chains for antihypertensive drugs in India's resource-limited public sector. Please find my comments, questions, and suggestions on the attached proof of the manuscript.

There were comments in track changes – response is given below:

1. Add secondary facilities in Intro – Abstract as highlighted in comment 3

Reply – Added – Abstract para 1

2. There are more recent surveys from India. The most recent one that I can think of is a statewide survey in 2020 from Kerala which reported low availability of several antihypertensive drugs.

Reply – Ref 8 added and text added on the Page 4, para 1, line 9-12

3. Do these include secondary care facilities too? Is there some data on the number of facilities at each level of care? 

Reply – Edits done - Page 5, Para 2.1, line 5

Every district has one district hospital and few districts also have sub district hospitals. We have not analysed separately because supply chain/ procurement is same irrespective of the facility. Also they are catering to relatively small proportion of patients among all enrolled patients. 

4. Explain morbidity based method

Reply – Details added – Page 6, Para 3 Forecasting drug requirements and budgeting – Line 6-8

The morbidity-based method takes into account expected patients to be initiated on treatment in addition to patients who are already on treatment and the drugs in the protocol at various levels.

5. Only for three quarters in the last year (2019-20)?

Reply – Page 9, bii and biii

Analysis is for one financial year. Clarified in the text

ii. The proportion of health facilities with drug stock out Out of all implementing facilities, the number of facilities had nil stock of any protocol drug at the end of each of the four quarters from Q2, 2019 to Q1, 2020. Stock out was separately considered for each of the three protocol drugs. 

iii. Drug stock in patient-days at the end of each quarter at the district level for the four quarters from Q2, 2019 to Q1, 2020.

6. Is this drug-specific or protocol-specific? If a facility did not have any unit of amlodipine, but stocked hydrochlorothiazide and telmisartan, would that not be a stockout? Suggest modifying the sentence to reflect that.

Reply – Page 9, bii Stockout is drug-specific. Sentence modified as below: 

ii. The proportion of health facilities with drug stock out Out of all implementing facilities, the number of facilities had nil stock of any protocol drugs at the end of each of the four quarters from Q2, 2019 to Q1, 2020. Stock out was separately considered for each of the three protocol drugs.

7. Is this assumption based on the early outcomes of IHCI? If yes, I suggest citing it here. https://www.nature.com/articles/s41371-022-00742-5

Reply – Ref 14 added. Assumptions are based on analysis of data from initial months. Overall results presented in the manuscript mentioned above. Recently we also did deeper analysis – stepwise for protocols, which is currently under review. Pre-print is available at medRxiv - 

Kaur P, Sakthivel M, Venkatasamy V, Jogewar P, Gill SS, Kunwar A, Sharma M, Pathni AK, Durgad K, Sahoo SK, Wankhede A. India Hypertension Control Initiative-Blood pressure control using drug and dose-specific standard treatment protocol at scale in Punjab and Maharashtra, India, 2022. medRxiv. 2023:2023-08.

8. Could you mention the years of issue of each EMLs here?

Reply – We do not have the information about the year for EML.

9. Are all IHCI protocols based on monotherapy with multiple pills? Is there any state protocol that uses single pill combinations?

Reply – The states included in the study did not have Fixed-dose combination antihypertensive drugs in the EML. Hence, it is not procured. Treatment protocols were decided in consultation with the state-level experts and program managers. They included drugs in EML in the protocols to enable timely procurements.

10. Not including the subsequent year (2020-21) appears to be a missed opportunity. I would be curious to know if (and how) the pandemic/lockdowns affected the IHCI procurement and implementation in general across various states, but perhaps beyond the scope of the current paper.

Reply – Data collection for drugs is extremely challenging. We established a system where monthly data collection was done involving pharmacists/nurses. During the pandemic, most staff were diverted to COVID-19 work, hence, we could not get complete data during that period. We have documented our experience of HT treatment access during the pandemic in the project districts in the following publication: 

Kunwar A, Durgad K, Kaur P, et al. Interventions to ensure the continuum of care for hypertension during the COVID-19 pandemic in five Indian states—India Hypertension Control Initiative. Global Heart 2021;16(1). 

11. The sentence appears unclear/incomplete. Suggest rewording.

Reply – Page 15, last 3 sentences.

We have modified the sentence - Initially, the other three states had to add chlorthalidone or hydrochlorothiazide in the EML, and procurement was initiated in the subsequent years.

12. Are these findings from the in-depth interviews?

Reply – Our team closely worked with the state and district-level officials for supportive supervision and documented the challenges. These observations were based on the real-time implementation experience. 

13. I believe this is Madhya Pradesh and not Maharashtra. If it's the latter, the next sentence is contradictory.

Reply – Agree – Correction done on page 17, para 3.5 – line 7

14. of telmisartan/losartan?

Reply – Added – Correction done on page 17, para 3.5 – line 7

15. Interesting to see Punjab having the highest % of facilities with stockouts for all three drugs in all quarters (except Diuretic in June 2019). Could this pattern be explained.

Reply – Punjab was the first state to start the IHCI project. Before IHCI, the procurement system was not geared up for NCD drugs. Although procurement/supply was improved, it could not keep up with increasing demand.

16. WHO HEARTS recommends 3-month supply.

Reply – We agree, and we hope eventually it will be possible. However, procurement and distribution systems are not mature enough for 3-month dispensing. We developed the operational guidelines in consultation with state NCD program managers. All states agreed for dispensing one month drugs and gradually moving to 2 or 3 months dispensing.

17. Weren't the IHCI protocols developed much before 2021? I would not agree that these protocols are consistent with the "recent" WHO guidelines (2021) as WHO strongly recommends initiation with combination therapy for most patients with hypertension (although algorithm-2 is monotherapy-based).

Reply - WHO guidelines do provide options for protocols starting with one drug. We used WHO HEARTS which also included protocols starting with drugs. Our experience with protocols starting with one drug better suited for low-resource settings informed the WHO guidelines. PI is part of the WHO guidelines review group. 

18. Figure 2 and 4 - Is this chlorthalidone/hydrochlorthiazide?

Reply: Correction done in the figures 2 and 4 

Initially, most states had HCTZ as there were no rate contracts for chlorthalidone. Since protocol had chlorthalidone, rate contracts were initiated and gradually, procurement transitioned from HCTZ to Chlorthalidone. 

Reviewer #2: The manuscript presents data and analysis that would be useful to readers, in particular health policymakers in India.

• My main concern is that the conclusion and discussion of this paper is not reflective of the findings. It seems to try to gloss over procurement concerns that the data actually reveal. For example, consider the abstract's conclusions that "This study demonstrates the feasibility of strengthening the procurement and supply chain management for antihypertensive drugs." The data don't exactly demonstrate this.

Reply: We agree with the reviewer's comments and have made the following changes in the abstract conclusions.

“As the program was rapidly growing, there were still gaps in the procurement and distribution system which needed to be addressed to ensure the adequacy of drugs.”

In addition, we highlighted the challenges that could not be addressed on Page 22 – last 3 sentences.

We are cautious about the balanced conclusions as there were major improvements in the supply chain and we hope this work will motivate the health departments to strengthen the systems further. We understand the limitations, however, system improvements take much longer, and our idea is to document the incremental changes possible despite tremendous challenges and limitations. 

• From Table 2, I calculated that the overall percent of facilities with stockouts actually increased over the study period for both Amlo and Telmisartan. So even though the absolute quantity of pills procured did grow substantially, there were some inefficiencies still. 

Reply: Page 17 – last para, Page 18 – First para

We agree with the reviewer’s observations. We have added sentences in the results – to highlight this issue. 

As the program was rapidly expanding, Punjab and Maharashtra had an increase in the facilities with stockout in the last quarter of 2019. 

In the first quarter of 2020, three states except Madhya Pradesh had more than 10% of facilities with stockouts for telmisartan/losartan. 

• According to figure 4, this seems to reflect issues with inadequate stock particularly of Telmisartan in 3 out of 4 examined states. For Amlo and Chlortalidone, issues with inadequate stock were observed in 1 out of 4 states.

Reply to comment:

We agree with the reviewer’s observations. We have mentioned this in the results and also in the conclusions. The reason was that pharmacists were not used to maintaining the 2-3 months stock because there was no system in place to assess the adequacy of stock. Since the supply chain is not tuned to maintaining such stocks, it will take longer to strengthen the processes. 

• Overall, the study demonstrates that the absolute quantities of pills did increase across the board, but some states still experienced inefficiencies with the sufficient procurement of some medicines, especially Telmisartan. The key conclusion of the paper should be to emphasize the need to resolve these stocking issues as scale up of the program increases. It's important to not bury the key take-away of the analysis just because it doesn't show perfect outcomes.

Reply : Page 24 – first para

We have modified the conclusions as per suggestions. 

Although the number of procured drugs increased, there were still insufficient stocks at various time points. It was due to the challenges which could not be fully addressed including timely tendering to ensure rate contracts, timely placement of orders, robust stock tracking system at the facility level, and timely availability of transport. It is important to address these gaps as the program expands and the number of patients on treatment increases.

---

## [Decision Letter · Decision Letter 1]

21 Nov 2023

Improving the availability of antihypertensive drugs in the India Hypertension Control Initiative, India, 2019-2020

PONE-D-23-13904R1

Dear Dr. Kaur,

We’re pleased to inform you that your manuscript has been judged scientifically suitable for publication and will be formally accepted for publication once it meets all outstanding technical requirements.

Kind regards,

Dzintars Gotham

Academic Editor

PLOS ONE

Additional Editor Comments (optional):

Reviewers' comments:

Reviewer's Responses to Questions

**Comments to the Author**

1. If the authors have adequately addressed your comments raised in a previous round of review and you feel that this manuscript is now acceptable for publication, you may indicate that here to bypass the “Comments to the Author” section, enter your conflict of interest statement in the “Confidential to Editor” section, and submit your "Accept" recommendation.

Reviewer #1: All comments have been addressed

Reviewer #2: All comments have been addressed

2. Is the manuscript technically sound, and do the data support the conclusions?

Reviewer #1: Yes

Reviewer #2: Yes

3. Has the statistical analysis been performed appropriately and rigorously? 

Reviewer #1: Yes

Reviewer #2: Yes

4. Have the authors made all data underlying the findings in their manuscript fully available?

Reviewer #1: Yes

Reviewer #2: (No Response)

5. Is the manuscript presented in an intelligible fashion and written in standard English?

Reviewer #1: Yes

Reviewer #2: Yes

6. Review Comments to the Author

Reviewer #1: (No Response)

Reviewer #2: The comments have been sufficiently addressed. For clarity, I recommend adjusting the first sentence of the conclusion from "This study demonstrates that including best practices can gradually strengthen the

procurement and supply chain for antihypertensives in a low-resource setting."

to

"This study demonstrates an increase in health system capacity to successfully procure increased quantities of antihypertensive medications commensurate with IHCI best practices."

7. PLOS authors have the option to publish the peer review history of their article (what does this mean?). If published, this will include your full peer review and any attached files.

Reviewer #1: **Yes: **Gautam Satheesh

Reviewer #2: No

---

## [Editor Report · Acceptance letter]

5 Dec 2023

PONE-D-23-13904R1 

Improving the availability of antihypertensive drugs in the India Hypertension Control Initiative, India, 2019-2020 

Dear Dr. Kaur:

I'm pleased to inform you that your manuscript has been deemed suitable for publication in PLOS ONE. Congratulations! Your manuscript is now with our production department. 

Kind regards, 

on behalf of

Dr. Dzintars Gotham 

Academic Editor

PLOS ONE